# The Meaning of Suicidal Behaviour for Portuguese Nursing Students

**DOI:** 10.3390/ijerph192114153

**Published:** 2022-10-29

**Authors:** Kelly Graziani Giacchero Vedana, José Carlos dos Santos, Tiago Carlos Zortea

**Affiliations:** 1Ribeirao Preto School of Nursing, University of São Paulo (EERP-USP), Sao Paulo 14040-902, Brazil; 2Health Sciences Research Unit, Nursing School of Coimbra, 7001, Bissaya Barreto Avenue, 3046-851 Coimbra, Portugal; 3The Oxford Institute of Clinical Psychology Training & Research, University of Oxford, Oxford OX1 3SZ, UK

**Keywords:** attempted suicide, suicide, nursing student, suicidology training

## Abstract

Background: The nursing perspectives on suicidal behaviors may influence the quality of assistance and suicidal prevention. This phenomenon is scarcely investigated among nursing students. Aims: The aim of this study is to understand the meanings of suicidal behavior for Portuguese undergraduate students. Methods: This qualitative study utilized Grounded Theory and Symbolic Interactionism. We collected data in Portugal in 2017–2018 with 13 undergraduate students. Results: Students compared suicidal behavior to “A complex and close haze” and considered it “A neglected phenomenon”. Suicidal behavior was predominantly perceived as an emotional distress that requires assistance. The students compared the person and society as “The car and the road: behavior influenced by communication and interaction” and valorized social dimensions and repercussions of suicidal behavior. Limitations: Lack of triangulation in the data and the sampling restricted to nursing students of a single institution are considered limitations of this study. Conclusions: This study can contribute to the development of academic education strategies and psychosocial support for nursing students.

## 1. Introduction

Suicide is a global concern that must be given priority in the public health and policy agenda. It is estimated that over 700 thousand people die by suicide every year c In Portugal, in 2019, the suicide mortality rate was 11.5 per 100,000 inhabitants, a higher rate than other types of violent deaths. These numbers are probably underestimated, as suicide is underreported in Portugal for several reasons, and the country is one of the countries with the highest number of violent deaths from undetermined causes in the European Union [1,2].

Suicidal behavior is defined as a set of thoughts and actions linked to the desire to cause one’s own death. It can be observed along a continuum that includes self-destructive thoughts, threats, gestures, and attempts to death by suicide [1,3]. In the literature, there are different theoretical models that seek to explain suicidal behavior and the complex interaction between biopsychosocial factors associated with this phenomenon. The most accepted models have in common the multifactorial perspective of suicidal behavior (associated with emotional, social, and physiological factors), the search to understand the complex mechanisms related to this behavior, and the transition from suicidal thoughts to suicidal actions [4,5,6]. The risk factors for suicidal behavior include previous suicide attempts, mental disorders, harmful use of psychoactive substances, stressful events, hopelessness, unbearable emotional suffering, feeling of failure, imprisonment, loneliness, lack of social support, accessibility to lethal means, exposure to suicide, violence, feelings of worthlessness, impulsiveness, aggression, among others [1,5,6]

Albeit it is argued that suicide can be prevented, the topic is hugely complex, still stigmatized, and poorly understood. These barriers ultimately increase the pain of those who suffer from suicidal thoughts and behaviors as well as prevent their families from seeking and obtaining effective and qualified assistance [1].

Qualitative studies carried out in Portugal reveal that suicide is mainly portrayed as an enemy, malefic, pathological, mysterious, and a threatening entity that causes high mortality in the Portuguese population and is associated with a public moral duty of prevention [7,8].

Preventive interventions must be comprehensive and involve multiple sectors of society [9]. Nursing professionals can play a crucial role in suicide prevention and provide significant insight into the prognosis of people at risk of suicide [10,11,12,13]. Suicidal behavior is considered an enigmatic phenomenon, and assistance to people with suicidal behavior is considered by nursing professionals as a critical and challenging moment, which evokes feelings varied and requires knowledge, skills, and emotional control [14,15,16].

Research reveals that nurses and nursing students often feel emotionally affected by suicidal behavior and have difficulties understanding, empathizing, and verbally interacting with those who attempted suicide [14,17,18,19]. When nursing professionals manifest judgments that lack empathy and do not feel prepared or supported for care, they may play a limited role in care (restricted to physical demands) [14].

Qualitative studies carried out in different countries show that the meanings attributed to suicidal behavior may converge or be dissonant with scientific advances in the understanding of suicide and its prevention. The meanings attributed to suicide are dynamically reconstructed, and it is common to use personal beliefs to support people with suicidal behavior [19,20]. A study carried out with Brazilian professionals revealed that they considered suicide as an instigating, unacceptable, and intolerable behavior [14]. A study carried out in Ghana suggests that suicidal behavior is reprehensible and objectionable, and its perceptions may favor moralistic attitudes and prescriptive approaches [21]. A Belgian study carried out on psychiatry specialist nurses identified different approaches, which may be focused on verifying and controlling the risk of suicide or understanding and connecting with the person [12]. In some contexts, it was identified the understanding of suicidal behavior is a health condition that requires care and interpersonal connection [13].

Studies show a predominance of negative attitudes toward suicidal patients and that these attitudes seem to be associated with a lack of appropriate training for health workers [22,23,24]. On the other hand, adequate training is associated with favorable changes in attitudes and competencies in assisting suicidal patients [25,26]. These findings stress the importance of qualified academic training for health professionals.

Studies indicate that skills and knowledge for suicide prevention have been insufficiently addressed in the academic environment. These gaps may favor the continuity of some negative beliefs and behaviors identified in society, such as judgments, discriminatory attitudes, lack of understanding, and the search for blame, admiration, or condemnation [18,27,28]. The representations of suicidal behavior among health students may impact peers’ support during the academic path, the help-seeking behaviors, and can also interfere with the quality of care provided to people with suicidal behavior [18]. 

Nursing care may be influenced by the nurse’s beliefs about how nursing professionals should proceed when working with suicidal patients. These perceptions are demonstrated to be associated with the training nurses received, and their skills in suicide risk assessment, and care planning [19,21,29,30]. However, the available evidence on what may shape nursing professionals’ perceptions of suicidal behavior is still scarce [31,32]. There seems to be a limited understanding of nursing students’ personal and professional experiences with suicidal behavior and the influence this may have on their learning [16,28]. Additionally, little is known regarding the educational content on suicide in undergraduate nursing curricula internationally [33], and the currently available studies on these issues are predominantly quantitative and restricted to suicide-related attitudes or specific components of professionals’ experiences, limiting a broader understanding of the representations of suicide among prospective nurses. In addition, there is a lack of studies conducted on nursing students. Knowledge of the meaning of suicide from the perspective of nursing students could shed light on the needs, potentials, facilitators, and limitations of academic training on suicidology, as well as the assessment of experiences and educational strategies related to suicide. Therefore, the aim of the present study is to investigate the meaning of suicidal behavior from the perspective of Portuguese nursing students.

## 2. Materials and Methods

The present study was designed to answer the following guiding question: What is the meaning of suicidal behavior from the perspective of Portuguese nursing students? The qualitative approach was adopted to meet the objective of this study as it is suitable for apprehending and interpreting representations, meanings, motives, beliefs, and values to obtain in-depth knowledge of different dimensions of social phenomena [34]. The study was presented in accordance with the Consolidated Criteria for Reporting Qualitative Research (COREQ), thereby ensuring the comprehensive and explicit reporting of the study [35]. 

The main strategies used to promote rigor in this research were: transparency in the description of the method; returning to the participants for validation when the accuracy/interpretation of interview transcripts was in question; rigorously and critically following Grounded Theory (GT) procedures; promoting regular discussions about both the reflexivity and credibility of the preliminary interpretations with a group of researchers; and an external audit of the research process [34,36,37].

In this study, we employed Symbolic Interactionism (SI) as a theoretical framework. SI provides the theoretical basis for our understanding of how meaning is developed through interaction. These theoretical assumptions guided the researcher’s perspective during the analysis of the phenomenon investigated. SI considers that behavior (observable external actions and internal experiences) is guided by the individual’s definitions of reality and that these definitions are derived from social interactions in which active individuals exert mutual influence [38]. Applying SI to the present investigation, suicidal behavior can be viewed as continually defined and redefined by nursing students through a dynamic and interactive interpretative process. 

We employed the version of GT proposed by Strauss and Corbin (2008) as the methodological framework of this study. We chose to use GT since it allows for the construction of a substantive theory with an emphasis placed on the social and psychological processes involved thereof (including meanings, perceptions, and how individuals continually reinterpret and react to the phenomenon). In this study, GT was used to determine methodological procedures such as theoretical sampling, memo writing, the constant comparative method, coding and categorizing, and theory generation [34]. SI is considered a component of the theoretical underpinnings of GT methodology [34] and the qualitative health research literature has reinforced the solidified conceptual linkage between SI and GT [39].

The study was conducted at the Nursing School of Coimbra, in Coimbra, Portugal. This higher education institution offers a degree course in nursing. Undergraduate students in nursing from the 5th semester onwards were eligible to participate in the study. The decision to approach the students in the final semesters was justified by the fact that they are most likely to have had contact with patients admitted to the hospital for having attempted suicide. 

Theoretical sampling was adopted to guide participant inclusion in the study according to their potential to describe experiences or their possible contributions to better understanding the investigated phenomenon. Initially, we used purposive sampling when inviting the first participant of the study, and further data were collected on the basis of theoretical sampling, which aims to maximize the opportunities to explore and compare events, concepts, characteristics, situations, experiences, and definitions, thus ensuring the constitution and refinement of study categories [34]. As the data were collected and analyzed, subsequent decisions about the methodological sampling of participants and the type of data collected were guided by the emerging theory. In this study, the interruption of data collection and the addition of new participants were determined by the theoretical saturation, which occurred when the objective of the study was reached, the categories of the study were developed, coherent, and articulated, and data became repetitive and added no relevant information for the understanding of the studied phenomenon. Additionally, for ethical reasons related to suicide prevention [40], we included in the study the number of students considered necessary and sufficient to achieve the proposed objective. During the theoretical sampling process, we invited 16 students to participate in the study. Three refused to participate due to a lack of availability. Thus, the research was developed with 13 students.

We collected data from November 2017 to February 2018 through individual, audio-recorded, open, semi-directed, or semi-structured interviews. The initial interview with the participants was guided by the following question: “Can you describe the meaning of suicidal behavior?” Other questions were subsequently added to clarify our analysis of their meaning of suicidal behavior. We continuously modified the interview process according to the analysis of the data obtained. A structured questionnaire was also employed to obtain demographic information (age, gender), and data relating to the participants’ academic backgrounds (semester of the undergraduate program, attending discipline on mental health, class, scientific events, courses or lectures on suicide prevention, contact with suicidal patients, and reading the literature on the subject).

The interviews were conducted in a private room at a time previously arranged with the participants according to their availability. All the interviews were conducted by a trained researcher (first author) who had no previous relationship with the participants, did not belong to the staff of the institution, and was not involved in educational activities. Each participant attended one or two interviews, approximately 40 min in duration. 

The transcribed interviews were analyzed through open, axial, and selective coding, in accordance with GT. We employed a constant comparative process for the identification of patterns and variations in the data, asking questions and sampling based on evolving theoretical concepts. Hypotheses about the emerging concepts and their relationships were developed and tested through the use of constant comparative data analysis [34]. The participants in the study, as indicated by GT, validated the results. 

During open coding, the data were broken into discrete parts, closely examined, and compared in terms of the similarities and differences exhibited between described events, situations, or relevant participant characteristics. Through axial coding, the categories were related to their subcategories, and the properties and dimensions of categories were refined to form precise explanations thereof. In selective coding, the categories were clearly integrated and refined; the theoretical scheme was reviewed to verify its internal consistency. Memos and diagrams were composed to support the development of the research. The research was conducted entirely in Portuguese and subsequently, the final report was translated into English.

The research began after receiving approval from the Research Ethics Committee of the Nursing School of Coimbra (ethical approval number: 377_12-2016).

We initially obtained a list of nursing students enrolled in the institution and invited them to confidentially participate in the study. Eligible participants were asked to take part in a study investigating the meaning of suicidal behavior. They were informed about the development and purposes of the study, and all participants provided written informed consent prior to their participation. We informed potential participants that their anonymity would be preserved, that they were free to refuse to participate in the study, would not be paid to participate, and that they could withdraw from their participation at any time without consequence. Considering the potential distress this could have upon the participants, the Informed Consent included a statement in which the researchers offered their support in the event of such an outcome. Although support was offered, none of the participants required it. The participants were also informed that in the educational institution there was a professional who could meet their emotional demands, if necessary. The work complied with all standards and recommendations concerning research involving humans.

## 3. Results

In total, 13 students participated in the study. Most participants were 22 years old (53.8%), women (92.3%), in the seventh semester of the undergraduate program (53.8%), had already attended a discipline on mental health (92.3%) and a course on suicide prevention (92.3%), stated they had been in contact with patients at risk of suicide (69.2%), had read literature on the subject (69.2%) and had not participated in scientific events, courses or lectures on the subject (76.9%).

Qualitative data analysis resulted in the following categories: “A complex and close ‘haze’”, “The car and the road: behavior influenced by communication and interaction” and “A neglected phenomenon”. These categories and their respective descriptions are presented below.

### 3.1. A Complex and Close “Haze” 

Participants described suicidal behavior as a complex and close phenomenon, compared to a haze in terms of the abstraction, boundaries, characteristics, and magnitude of this behavior. The haze was a metaphor for the psychological state of suicidality, which includes overwhelming and permanent suffering, hopelessness, loss of self-orientation, and a dysfunctional way to handle a desperate situation.

“It’s like a foggy day, where you can’t see the blue of the sky. And the person stops believing skies exists, that the blue in the sky is real.” (P5, 2018)

The respondents considered suicidal behavior complex, intriguing, multi-causal, with imprecise boundaries, and marked by uniqueness, misunderstandings, and controversies. Students’ views on suicidal behavior seemed obscure or had not been thought about or discussed before.

“There are predisposing factors, stress factors, biological factors […] it’s a mixture of things, it’s not just one thing.” (P11, 2018)

“Suicidal behavior can have many reasons, they are variables […] I can’t give you a specific definition, they are actions intended to hurt themselves.” (P2, 2017)

“I never thought about it, but in my view, suicide is always different because of the people, because each person is different.” (P10, 2018)

Suicidal behavior was also considered common in society, something that can occur at any stage in life, associated with regular, everyday pressures and hardships, and present in the social circle of many respondents.

“I had a schizophrenic uncle, and last year he committed suicide. And two years ago, an aunt with the same problem committed suicide and my grandmother attempted suicide […] suicide is a major part of my life.” (P1, 2017)

“I think it’s rare to meet people who don’t know someone who attempted [suicide].” (P3, 2017)

### 3.2. “The Car and the Road”: An Interactional and Communicative Phenomenon

Suicidal behavior was considered essentially interactional and communicative. It was not seen by the students as an isolated or individual event. Social dimensions were strongly present in representations of the risk factors, protective factors, and effects or consequences of suicide. A metaphor of a car (representing the individual) and a road (representing the social interactions at both a macro level, i.e., society, and a micro level or personal relationships) was used to describe the social dimensions of well-being or suicidal behavior.

“It’s like I [person with suicidal behavior] was a car and the people who support me were the road. I am the one who is in movement, but if the road goes away, I fall.” (P5, 2018)

“We do not live alone in the world; we have the people around us, who affect us in several ways, including this one.” (P3, 2017)

The statements revealed the belief that suicidal behavior and communication or interaction could exert a mutual and heavily influence in terms of prevention and support (information, suicide risk assessment, support, treatment, and respect), neglect (non-recognition of risk, passivity) or pro-suicidal effects (inadequate communication, increased suffering and risk factors, contagion).

“They [closer people] will know better how the person [vulnerable to suicide] is and what they need […] they can help this person better.” (P5, 2018)

“Sometimes we are only looking at ourselves and we forget about the others, we are not there for them, we stop thinking and knowing what we should do [to help people who are suffering].” (P8, 2018)

“In fact, the media can influence people. The media influences everything and everyone. […] people can read and get this idea for themselves.” (P3, 2017)

The students believed that the person who suffers from suicidal thinking and behaviors, intentionally or not, could be sending messages, “a cry for help” and “a cry for attention”, or trying to provoke feelings or influence others.

“One person can make a cry for attention just to get support or as a cry for help”. (P11, 2018)

“It’s selfish, with the intent to hurt others. […] they are attention-seeking behaviors.” (P12, 2018)

“When she is thinking about suicide, she is thinking: ’m going to relieve the burden, the weight that I mean for other people. If people get rid of me, they will be happier. I won’t be missed here.” (P5, 2018)

The consequences of suicidal behavior were also deeply linked to social dimensions that include the feelings, reactions, judgments, and impact on the history of lives. Death by suicide can have a profound effect on the people who are close to the deceased. Respondents who knew someone close who died by suicide claimed suicide leaves “permanent marks” and “affects mainly the people who remain”. 

“When there is a suicide, people usually analyze the social context involving the person.” (P7, 2018)

“I feel that suicide doesn´t happen to the person who dies, it happens to the other people (…) he or she stops feeling anything, while great suffering begins for the bereaved people.” (P1, 2017)

### 3.3. A Neglected Phenomenon

Suicidal behavior was considered a difficult phenomenon to identify and prevent effectively in the present moment. According to the participants, these difficulties were related to the characteristics of the psychological states of suicidality, and to society’s responses to this phenomenon. Thus, prevailed in society, reactive actions regarding the suicidal behavior (responses after the manifestation of the suicidal behavior) than proactive actions (early prevention, anticipation of actions with initiative).

“When it happens, the person is not expecting it. […] They only realize they could have done something after it already occurred.” (P9, 2018)

The difficulties with identifying a suicide state and, therefore, being unable to prevent it were the covert, non-specific, or not evident symptoms, the recurrence of crises with unforeseeable consequences, lack of hope and assistance to seek help, and the possibility of an abrupt suicide attempt.

“Suicide is a process that begins with negative thoughts, and it can go unnoticed by people who are not very close or attentive […] and it may be recurrent […] and when we’re at that stage we don´t want anyone’s help.” (P5, 2018)

“ […] It is difficult to distinguish things and we cannot always see clearly if a person is suffering. Some think there are fewer better days, because fewer good days exist and we neglect them, but sometimes I think it is very hard to see.” (P3, 2017)

The limited proactivity of society in preventing suicide could also be related to ignorance, lack of preparation and difficulties approaching the subject, stigma, discrimination, and condemnatory attitudes towards suicidal behavior. Participants also mentioned the generalization and trivializing of other people’s suffering, the conventional and impersonal way of supporting them, a limited bond with the person at risk, and personal attitudes or beliefs. 

“Just yesterday I had a meeting with a lecturer and with my colleagues. There was this colleague of mine, and we started talking about it [suicide], but he didn’t feel comfortable so we chose to remain silent because no one knew what to say.” (P11, 2018)

“I think that the prevailing concept, in general, is, for example: “He´s just sad”, “You have to be strong to get through this”. I think it’s something along those lines. […] they don’t pay attention, they neglect it, because they know other people have been there and overcame the situation.” (P12, 2018)

According to the respondents, suicidal behavior was an attenuated or deconstructing taboo in society. However, participants say that the demystification process is still superficial, insufficient to promote changes, and the discussion about suicide is yet restricted to certain groups. The social responses to suicidal behavior were associated with geographic locations (with distinct cultural aspects), level of education, access to knowledge about mental health, social isolation, and personality factors (especially openness and flexibility).

“I think it’s not discussed very much. […] but my generation, I think they have a new approach. It is becoming more and more common and easier to talk about.” (P2, 2017)

“The end of life is a sensitive topic, and in Portugal they [health professionals] still try to keep the person alive, regardless of the circumstances.” (P7, 2017)

“There’s the social stigma. What they say in the villages and out there is that the person who commits suicide is weak […] my mother is from a village and she would say psychologists are for crazy people […]. In the village, people are less literate, they rarely go to the city and their only means of communication are the TV and the radio, they are very isolated inside themselves.” (P5, 2018)

## 4. Discussion

The current study aimed to investigate the meaning of suicidal behavior from the perspective of Portuguese nursing students. Suicidal behavior was considered complex and diverse, intriguing, and multifactorial. It was represented as a neglected risk due to its characteristics (involving hidden emotional states and behaviors considered non-specific, abrupt, or recurrent) and social issues that impair suicide prevention (negligence, non-recognition of risk, communication inappropriate, and lack of proactivity). Suicidal behavior was considered essentially interactional and communicative. Individual, relational and social dimensions were present in risk and protection factors, in prevention, functions, intentionality, and consequences of suicidal behavior. The most accepted theoretical models regarding suicidal behavior have a multifactorial perspective and reinforce the relevance of individual, interactional and communicative dimensions [4,5,6].

In this study, suicidal behavior was perceived by the participants as a complex, but a common and neglected phenomenon. Previous research suggests that suicide is frequent in Portuguese society although notoriously underreported [7,8]. The silence around suicide in Portugal can limit the legitimacy, visibility, social relevance, and political discussion on the subject, which favors the perpetuation of inequities, stigma, and ignorance [7].

In this study, the students had difficulties describing suicidal behavior, stated by themselves, still as something obscure. At the same time, they also know they will be working with this uncommon phenomenon whose understanding is limited. In the literature, suicide is often described as a complex phenomenon with different connotations, hardly comparable to other potentially fatal conditions, and overshadowed by the lack of a clear and publicly stated definition [14,19,21].

Previous studies revealed that many health professionals, students, and a large proportion of the general population believe that people suffering from suicidal thinking and behavior are responsible and aware of the consequences of their suicidal actions. In addition, suicidal behavior is often associated with stigmatized and negative attitudes, and misunderstanding and is considered a transgression or reprehensible act [14,18,21,23,30]. In our sample, differently from other studies, the legitimation of emotional distress as a condition requiring care prevailed over the perception of suicidal behavior as a transgression of health care. Although different perspectives coexisted, it is possible to identify a pattern in the perception of suicidal behavior as a dysfunctional way of dealing with despair and suffering, and the students showed interest, empathy, and efforts to understand what someone experiences when they feel suicidal.

In another study with nursing students, suicidal behavior was considered an act that could be communicative when associated with manipulation or a request for help [18]. In the current study, suicidal behavior was essentially interactive and communicative. The relational and social aspects were considered in the evaluation of risk factors and protection, prevention, functions, intentionality, and the consequences of suicidal behavior. These perceptions may be influenced by academic training, but this result may also reveal individual experiences related to social support and mental health during the undergrad period Understanding adequately the social dimension of suicidal behavior and mental health issues is important for nursing care planning, social support and to avoid misunderstandings, blame or self-blame related to suicidal behavior in others. Appropriate pedagogy and student support services must be considered for nursing undergraduate students as the emotional intensity of dealing with suicide prevention is a focal point in preparing and supporting prospective nursing professionals [16,23,28]. A review of qualitative studies indicates that it is necessary to improve nurses’ relational skills and monitor the emotional impact related to suicide in order to promote more qualified care [15].

According to nursing students, the quality of social interactions could be related to prevention, neglect, or pro-suicidal effects. They vehemently criticized the predominance of reactive rather than proactive (preventive) attitudes related to suicidal behavior. Nursing care for people with suicidal behavior demands interaction, engagement, and connection with the person, understanding them, and establishing a therapeutic bond. This bond needs to have authenticity, trust, reflexivity, proximity, empathic assessment, and construction of collaborative care capable of favoring openness, building a support network, and better coping with crises [11,13].

Regarding the reactions to suicide, studies with students and health professionals point out that suicide can provoke reactions of emotional distress, concerns, doubts, fear, frustration, and shock [14,17,18,19], but it can also favor learning and proactive actions related to better risk management of suicide in clinical practice and personal determination to prevent suicide [27]. 

Studies indicate that suicide risk can be assessed by nursing students based on personal judgments and beliefs [19,20]. In addition, in different countries, health professionals also report difficulties in assessing suicide risk and developing appropriate interventions [12,14,21]. In our study, the students attributed the difficulties of recognition and prevention to the characteristics of the psychological states of suicidality (symptoms, recurrence, and unforeseeable consequences), and society’s responses to this phenomenon (unprepared to support, stigma, discrimination, condemnatory attitudes, generalization and trivializing the suffering, fragile or superficial social relationships). The use of standardized patient sessions, case studies, simulations, blending theory, role play, sharing of personal experiences of suicide, and reflexive approaches in training has shown promising results in the improvement of professional competences [16,26,28,33]. 

The students believed that the taboo surrounding suicide is in a process of deconstruction, but they also pointed out that the ongoing demystification is slow because people avoid talking about suicide. They also stated that the younger generation was more understanding than the previous ones. According to the Directorate General of Health in Portugal, stigma, and taboos related to suicide are still important problems in Portugal [2,8]. However, the literature indicates that open communication about suicide is fundamental for the evaluation of risk and protective factors, and for the development of preventive strategies [1,11,13]. Studies have suggested that the taboo and avoiding talking about suicide may also be prominent in the academic environment [18,27,28] and there are contexts in which nursing students controversially criticize taboos related to suicidal behavior, and reinforce by judgments, discriminatory attitudes, and avoiding talking about this subject (including in clinical practice) [18,27,28]. In our study, the undergraduate students also pointed out that the social stigma towards suicide and mental health issues is associated with geographic locations, level of education, and social isolation. These issues could be addressed in future studies in Portugal.

The limitations of this study involve a lack of triangulation in the data, the small sample size, and the sampling restricted to nursing students of a single educational institution within a restricted geographic territory. However, investigations that use GT can be adapted and expanded, from the emergence of new data or new sample groups. To our knowledge, this is the first study to address the perceived meanings of suicidal behavior amongst Portuguese nursing students.

## 5. Conclusions

Suicidal behavior was predominantly represented as emotional distress that requires assistance, and it was considered diverse, intriguing, and multifactorial. It was considered essentially interactional and communicative. Individual, relational, and social dimensions were present in risk and protection factors, in prevention, functions, intentionality, and consequences of suicidal behavior.

Suicidal behavior was perceived as a neglected risk due to its characteristics (involving hidden emotional states and behaviors considered non-specific, abrupt, or recurrent) and social issues that impair suicide prevention (negligence, non-recognition of risk, communication inappropriate, and lack of proactivity). 

It is relevant to investigate the elements that can harm prevention (taboos, stigma, ignorance, unpreparedness, generalization, and trivialization of the suffering of others) among other health professional students. Additionally, it is important to promote deep and open discussion about suicide and investigate the cultural issues and other characteristics (geographic locations, level of education, and social isolation) apparently related to stigma and taboos regarding suicide in Portugal.

## Data Availability

The data presented in this study are not available due to ethical restrictions.

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
