# Peer review of "The Meaning of Suicidal Behaviour for Portuguese Nursing Students"

_ijerph, 2022, doi:10.3390/ijerph192114153_

Round 1

Reviewer 1 Report

Nurses have a key role in the treatment of Suicidal behavior. But their role is underestimated and understudied, I think the authors wrote a very good and nice introduction on the current status of the field. However, the design and data collection do not convincingly help us to understand this process better. 

The qualitative design is based on grounded theory and social interaction. At the end, 13 nurses trainees participated in the study 16 were approached. It is a very specific group from one institution in a city in Portugal. With such a complex topic you cannot reach saturation with 13 persons. Also, there is a lack of controls. We can not rule out that trainees in other fields (or youth in this age range) would have the same opinion. 

The authors did a good job in theory and presenting the data available, but no specific conclusions should be made regarding programs for nurses. Awareness training about suicide is proven to be a preventive measure for all kinds of populations (police, military students). The Design lack the potential to really grasp the attitude (or difference in attitude) of nurses trainees

Author Response

Dear Reviewer,

We would like to thank you and the reviewers for yours comments. We are sure they will help to significantly improve the comprehension and the impact of the article. We reviewed the manuscript according to reviewers comments. We are honored for the opportunity of having our article evaluated and considered for publication on the International Journal of Environmental Research and Public Health.

REVIEWER 1

REVIEWER 1- COMMENT 1. “Nurses have a key role in the treatment of Suicidal behavior. But their role is underestimated and understudied, I think the authors wrote a very good and nice introduction on the current status of the field. However, the design and data collection do not convincingly help us to understand this process better. 

The qualitative design is based on grounded theory and social interaction. At the end, 13 nurses trainees participated in the study 16 were approached. It is a very specific group from one institution in a city in Portugal. With such a complex topic you cannot reach saturation with 13 persons. Also, there is a lack of controls. We can not rule out that trainees in other fields (or youth in this age range) would have the same opinion.” 

AUTHORS RESPONSE 1:

Thank you for the careful review and for the suggestions that give us the opportunity to improve our article and clarify aspects that have been pointed out.

This research, as we have already described in the article, was based on Grounded Theory and Symbolic Interactionism (SI), not “social interaction”, as you wrote. SI is traditionally identified as a component of the theoretical foundations of the GT version used in this study (Strauss & Corbin, 2008).  The integration of SI and GT is supported by several authors and widely used in rigorous scientific studies.

According to the Methodological Framework used, it would not be possible to state that "you can't reach saturation with 13 people". The sampling proposed by the GT is not evaluated numerically, but by the representativeness and theoretical saturation of the categories, criteria duly followed and presented in the study: “Theoretical sampling was adopted to guide participant inclusion in the study according to their potential to describe experiences or their possible contributions to better under-standing the investigated phenomenon. Initially we used purposive sampling when inviting the first participant of the study, further data being collected on the basis of a theoretical sampling, that aims maximizing opportunities to explore and compare events, concepts, characteristics, situations, experiences and definitions, thus ensuring the constitution and refinement of study categories (33). As the data was collected and analysed, subsequent decisions about the methodological sampling of participants and the type of data collected were guided by the emerging theory. In this study, the interruption of data collection and the addition of new participants were determined by the theoretical sat-uration, which occurred when the objective of the study was reached, the categories of the study were developed, coherent and articulated, and data became repetitive and added no relevant information for the understanding of the studied phenomenon”.

Additionally, for ethical reasons related to suicide prevention (Lakeman & Fitzgerald, 2009), in the sampling process, included in the study the number of students considered necessary and sufficient to achieve the proposed objective.

The inclusion of controls would not be mandatory (and could be controversial) to achieve the objective proposed in this study. We agree that “we cannot rule out that this opinion is found in other groups”. However, these "other groups" are not included in the objective of our study. In studies that employ GT, the theories generated can be deepened and expanded in future studies, by the inclusion of new sample groups.

Lakeman, R., & Fitzgerald, M. (2009). Ethical suicide research: A survey of researchers. International Journal of Mental Health Nursing, 18(1), 10–17. https://doi.org/10.1111/j.1447-0349.2008.00569.x

Strauss, A., & Corbin, J. (2008). [Basics of Qualitative Research: Techniques and Procedures for Developing Grounded Theory] (2nd ed.). Porto Alegre: Artmed.

REVIEWER 1- COMMENT 2. “The authors did a good job in theory and presenting the data available, but no specific conclusions should be made regarding programs for nurses. Awareness training about suicide is proven to be a preventive measure for all kinds of populations (police, military students). The Design lack the potential to really grasp the attitude (or difference in attitude) of nurses trainees.”

AUTHORS RESPONSE 2:

Thank you for this important suggestion.

We rewrote our conclusion to make it clearer and to avoid considerations that are not directly related to our results.

The authors

Reviewer 2 Report

The paper addresses an important global issue, which is not just important to the health caring professions, but more generally to people who support people who show suicidal behaviours and ideation. 

The authors spend quite a bit of space justifying their study with (mostly dated) reference to studies that talk about nurse attitudes to suicide. The problem with this is that they don't actually say what the attitudes reported in other studies are. It would seem obvious to me, given the research question, to unpack what studies say is the 'meaning' of suicide to nurses/students.

Then, given that your discussion goes on to talk about the implications of your findings, it would again make good sense to include literature on what instutions are doing in response to these attitudes, and whether there is any evidence that these actions make a difference. 

As it is, a considerable amount of new literature is introduced in the discussion, and so it appears to come out of nowhere, disconnected from the literature and seemingly randomly selected in response to the opinions of the authors.

I would also like to see a clearer and succinct response to the research question in the discussion. As it is, the authors have just parroted back the thematic headings that they have induced from their analysis.

In summary, some ways the paper could be improved:

1. update the literature. It's way too old.

2. include literature up front about what the meaning of suicide is to nurses and students.

3. include literature up front that shows the implications of these.

4. succinclty respond to your research question, in the discussion.

5. reflect critically on the literature and your findings in your discussion.

Author Response

Dear Reviewer,

We would like to thank you and the reviewers for yours comments. We are sure they will help to significantly improve the comprehension and the impact of the article. We reviewed the manuscript according to reviewers comments. We are honored for the opportunity of having our article evaluated and considered for publication on the International Journal of Environmental Research and Public Health.

REVIEWER 2

REVIEWER 2- COMMENT 1: “The paper addresses an important global issue, which is not just important to the health caring professions, but more generally to people who support people who show suicidal behaviours and ideation. 

The authors spend quite a bit of space justifying their study with (mostly dated) reference to studies that talk about nurse attitudes to suicide. The problem with this is that they don't actually say what the attitudes reported in other studies are. It would seem obvious to me, given the research question, to unpack what studies say is the 'meaning' of suicide to nurses/students.

Then, given that your discussion goes on to talk about the implications of your findings, it would again make good sense to include literature on what instutions are doing in response to these attitudes, and whether there is any evidence that these actions make a difference. 

As it is, a considerable amount of new literature is introduced in the discussion, and so it appears to come out of nowhere, disconnected from the literature and seemingly randomly selected in response to the opinions of the authors.”

(…)

“In summary, some ways the paper could be improved:

  1. update the literature. It's way too old.
  2. include literature up front about what the meaning of suicide is to nurses and students.
  3. include literature up front that shows the implications of these”.

AUTHORS RESPONSE 1:

Thank you for the careful review and for the suggestions that give us the opportunity to improve our manuscript. We update the literature and we improved the introduction and discussion according with the suggestions.

REVIEWER 2- COMMENT 2:” I would also like to see a clearer and succinct response to the research question in the discussion. As it is, the authors have just parroted back the thematic headings that they have induced from their analysis”.

“(…)

  1. succinclty respond to your research question, in the discussion.
  2. reflect critically on the literature and your findings in your discussion.”

AUTHORS RESPONSE 1:

We attended to these suggestions.

Round 2

Reviewer 1 Report

Although the authors did made some important changes. I do not agree that in Grounded therory a sample of 13 is enought to reach saturation. Als the data do not convince me that satuaration was reached Leaving a small sample with anactodical evidence which seems very hard to replicate

Author Response

We respect your opinion, although we disagree with it for the various reasons (based on the Grouded Theory) already mentioned in the previous response letter. We highlight that, for ethical reasons related to suicide prevention (Lakeman & Fitzgerald, 2009), in the sampling process, included in the study the number of students considered necessary and sufficient to achieve the proposed objective.

Reviewer 2 Report

I note the significant revisions the authors have made to this manuscript. Please see additional comments on the attached file.

In summary though, I think there is still room for improvement in this paper before publication. 

While I appreciate this study is based in Portugal and much of the literature is Portuguese, I think you could place your findings within an international context. You have adopted a 'victim blaming' definition and description of suicidal behaviour which is definitely not reflected in the international literature. What role does mental ill-health have to play? What are the reported stats on suicide in Portugal? What policies and strategies are in place to prevent suicide? What are the multiple causes of suicide?

You talk about the taboos of discussing suicide. Maybe this is true of Portugal but internationally the academic literature is definitely destigmatising discussions of suicide and I think this is particularly true of the health caring professions. A comparison with international studies might reveal similarities and differences that would give you the ability to make stronger conclusions.

I previously commented on the introduction of new literature in the discussion. This version does the same! ITs fine to discuss the literature but why wasnt it all discussed first in the literature?

My critique here should allow you to sharpen the paper and give it stronger conclusions.

Author Response

We carefully attended these suggestions. Thank you for all these comments and for sending the file with detailed suggestions related to different parts of the manuscript. This helped us to improve our work.

Now, all the references used at the discussion are previous presented at the introduction.

We place our findings in an international context. We clarified which considerations came from qualitative studies (from different countries) and how they approximate or not to theoretical models widely accepted in the international scientific literature.

We answer the questions raised by means of improvements in the manuscript.

The authors